# Identification of novel plasma proteomic biomarkers of Dupuytren disease

Blake Hummer[1], Paola Sebastiani[2,3], Anastasia Leshchyk[4], Anastasia Gurinovich[2,3], Cecilie Bager[5], Morten Karsdal[5], Signe Nielsen[5], Charles Eaton[1]*

1 Dupuytren Research Group, West Palm Beach, Florida, United States of America, 2 Institute for Clinical Research and Health Policy Studies, Tufts Medical Center, Boston, Massachusetts, United States of America, 3 Department of Medicine, Tufts University, Boston, Massachusetts, United States of America, 4 Department of Medicine, Computational Biomedicine Section, Chobanian and Avedisian School of Medicine, Boston University, Boston, Massachusetts, United States of America, 5 Nordic Bioscience, Herlev, Denmark

☉ These authors contributed equally to this work.
* c.eaton.md@dupuytrens.org

## Abstract

Dupuytren Disease (DD) is a chronic progressive disease that can cause disabling hand deformities. The most common treatments have either high complication rates or high early recurrence rates. Dupuytren lacks a staging biomarker profile to inform the development of preventive therapeutics to improve long-term outcomes. This multi-omic study aimed to create a DD blood proteomic biomarker profile by comparing DD plasma with that of a healthy control group. We measured circulating collagen metabolism peptides and found normal Collagen I synthesis but impaired Collagen I degradation in DD. We measured 6995 serum protein aptamers and identified 68 proteins that showed statistically significant differences compared with the control group. We developed two Diagnostic Proteomic Risk Scores (DPRS) based on hypothesis-free and hypothesis-based analyses. In independent data, our hypothesis-free and hypothesis-based DPRS distinguished Dupuytren from control subjects with accuracies of 76.5% and 70.6%, respectively. Our hypothesis-based DPRS also distinguished DD subjects with different disease progression rates by age at their first corrective procedure (p = 0.0018). This pilot study is the first to provide evidence to suggest that Collagen I accumulation in DD results from impaired degradation rather than increased collagen synthesis. It also describes novel DPRS that have potential use as diagnostic and staging biomarker panels for Dupuytren disease.

## Introduction

Dupuytren disease (DD) is a progressive fibrotic condition that often results in finger deformities called Dupuytren contracture (DC). DD affects an estimated 10 million Americans. DC is associated with loss of productivity, higher healthcare costs, and a greater prevalence of benign diseases [1].

**Data availability statement:** All the data and analytic scripts necessary to replicate the findings are available in the GitHub repository Dup_Blood_Proteomic_Data at https://github.com/Dupuytren/Dup_Blood_Proteomic_Data.

**Funding:** The author(s) received no specific funding for this work.

**Competing interests:** The authors declare the following competing interests: Morten Karsdal, Cecilie Bager, and Signe Holm Nielsen are employees of Nordic Bioscience and hold stocks in Nordic Bioscience. No other competing interests have been declared. This does not alter the authors' adherence to PLOS ONE policies on sharing data and materials. The rest of the authors have no conflicts of interest to declare.

The greatest need in DD care is preventing DC progression and recurrence. As with other fibrotic diseases, developing preventive treatments requires a biomarker profile for staging and measuring therapeutic response. The authors have not found published reports with statistically significant evidence of preventing DC recurrence or a biomarker tool to develop such treatments. This pilot study aimed to develop preliminary data on a diagnostic blood biomarker profile of DD.

DC is a common outcome of DD. Of those diagnosed with DD at age 65, one in four men and one in eight women will require at least one corrective procedure over their lifetime [2]. The most common current treatments are palliative, often resulting in partial or temporary improvement [3]. Surgery for DC has a greater risk of major complications than other common elective hand operations [4–7], and recurrent DC is the most common diagnosis leading to elective finger amputation [8].

Genetic factors account for about 80% of the risk of developing DD [9]. Family history, age, and gender are strongly associated. A genetic risk score (GRS) based on 11 SNPs correlates with DD diagnosis [10], and a weighted GRS based on 26 SNPs correlates with post-procedure recurrence [11].

DD affects the palm skin, subcutaneous tissues, and fascia [12]. Early disease is cellular, with fibroblasts, myofibroblasts, and perivascular clusters of myofibroblasts, pericytes, and mononuclear immune cells [13]. Early changes include local variations of immune [13], hematopoietic, and mesenchymal cell markers [12], cytokines, signaling factors [14], and extracellular matrix components [15,16]. Progression transforms tissues into nearly acellular, densely packed collagen.

Physical changes are the basis for DD diagnosis, staging, and treatment, but physical findings do not predict progression [17]. There are no standard noninvasive tests of DD biological activity. Biopsy is problematic because it can trigger disease activity and progression. Blood remains the most promising source for a disease activity profile.

## Materials and methods

### Study approval

The Institutional Review Board Ascension Via Christi Hospitals, Wichita, Inc., approved this study and the associated informed consent form. Written informed consent was obtained from all participants before their participation. This written informed consent included a description of the use of the photographs/ images to be reviewed for study entry. The records of these signed informed consents are held on file by the investigators in accordance with PLOS policies. All samples remained blinded throughout the processing and data collection in the laboratories. All methods were performed in accordance with relevant guidelines, regulations, and the Declaration of Helsinki.

### Subjects

We selected disease cohort subjects from the International Dupuytren Databank (IDDB), a patient-reported registry developed by the Dupuytren Research Group

(DRG) in collaboration with the National Databank of Rheumatic Diseases (NDB). Control subjects were enrolled by word of mouth and from the NDB database (https://dupuytrens.org/enroll-in-the-iddb-2/). Study candidate subjects were identified on December 3, 2019, using the entry criteria described below. We excluded minors from the study. Our study involved male and female subjects. Sex was considered a biological variable. While we controlled for sex differences, we did not stratify by sex due to limited sample sizes. The authors had no access to information that could identify individual participants during or after data collection. All subjects met the entry criteria, and each subject submitted a signed physical Informed Consent Form by postal mail. Recruitment started on December 13, 2019, and ended on May 3, 2021.

## Entry criteria

The DD cohort met criteria based on clinical evidence of DD with strong traditional predictors of recurrent deformity after a corrective procedure: age of disease onset younger than 50; two or more first-degree relatives with DD; current DD nodules; three or more separate locations of DD, or two palm locations and Ledderhose disease (a related disorder). Control cohort subjects were healthy individuals with no clinical evidence of DD, no personal history of DD or Ledderhose disease, and no family history of DD.

We confirmed study eligibility by requiring candidates to upload recent images of the palmar and dorsal sides of each hand. The Principal Investigator/senior author confirmed the entry criteria and reviewed images to confirm clinical evidence of active DD in the disease cohort and the absence of these findings in the control cohort.

## Blood sample collection

The study used a mobile phlebotomy service (Phlebotek Solutions Corporation, Fort Lauderdale, FL) to collect blood specimens. Once confirmed that the subject met all study entry criteria, they were contacted by DRG and later by Phlebotek to schedule their blood draw. The phlebotomist visited the subjects in their homes and collected two tubes of blood. The last 16 blood draws included a third 2.5 ml PAXgene tube collection for additional genomic testing.

Processing of the EDTA tubes for proteomic testing was standardized. Immediately following collection, blood samples were centrifuged for 15 minutes at 2000 g to generate plasma. The plasma was harvested from the top of the container and pooled before aliquoting into storage cryotubes. The last 500 μL of plasma was discarded to avoid contamination with platelets. Samples were immediately packaged for shipping on dry ice.

## Number of subjects

One hundred subjects were planned (50 DD/50 Control). This number was a practical consideration, given the limited number of potential participants enrolled in the International Dupuytren Data Bank who met all disease cohort entry criteria. Blood collection launched in February 2020, just as the COVID pandemic began. Due to COVID-related issues, we amended the protocol to enroll 50 subjects. Forty-seven subjects were enrolled; 46 subjects had samples drawn. One sample was disqualified due to failed shipping refrigeration. Mass spectrometry (Bioproximity, Manassas, VA) was performed on samples from the remaining 45 subjects (25 DD, 20 Control). Twenty-seven subjects (13 DD, 14 control) had second blood samples drawn 6–14 months after their first draw for mass spectrometry testing by a second lab (MacCoss Lab, Seattle, WA). Additional proteomic testing was conducted on remaining aliquots: 1) n = 27 redrawn samples had ELISA tests for collagen metabolites (Nordic Bioscience, Herlev, DK); 2) n = 72 from both initial and redraw specimens had Aptamer-based testing (SomaLogic, Boulder, CO). S1 Fig in S1 File outlines the distribution of samples and laboratories.

## Blinding

All subjects were assigned and identified by participant numbers. Investigators were blinded to the sample group allocation. All samples remained blinded throughout the processing and data collection in the laboratories.

 

## Assays

No laboratory protocols were conducted in-house.

**Nordic Bioscience collagen markers.** Circulating collagen synthesis and degradation markers were measured using proprietary assays [18]. Synthesis biomarkers included collagens I (PRO-C1), II (PRO-C2), III (PRO-C3), IV (PRO-C4), V (PRO-C5), VI (PRO-C6), VII (PRO-C7), and IX (PRO-C9). Circulating degradation markers were measured for collagens I (C1M), II (CALC2, C2M), III (C3M, CTX-III), IV (C4M), VI (C6M), and VII (C7M). All biomarkers were assessed by competitive ELISAs and validated to measure in human plasma samples. The inter- and intra-assay coefficients of variation were >15% and 10%, respectively, for all assays.

**Mass spectrometry.** A novel untargeted proteomics assay using large-scale liquid chromatography-tandem mass spectrometry (LC-MS/MS) with data-independent acquisition (DIA) was used to collect proteomic data. Each sample (10 μl plasma), following tryptic digestion, was diluted 50:50 by the addition of 4% sodium dodecyl sulfate (SDS) buffer to a final concentration of ~2% SDS. Approximately 200 μg of protein was added to 96-well plates, and the remainder of the sample preparation was performed using the single-pot, solid-phase enhanced sample preparation for proteomics methodology (SP3) [19,20]. The entire process of protein capture, reduction and alkylation, protein cleanup, digestion, and peptide cleanup was performed using a KingFisher magnetic bead robot [20]. Peptides were reconstituted in HPLC starting aqueous buffer (5% acetonitrile and 0.1% formic acid) with analysis performed on the high-resolution quadrupole-orbitrap hybrid mass spectrometer (Exploris 480, Orbitrap Fusion Lumos, or Orbitrap Fusion Eclipse).

Data was acquired in a single, 2-hour LC-MS/MS run (90 min HPLC gradient) on a Thermo Quadrupole-Orbitrap mass spectrometer. Crucially, the DIA method comprehensively sampled all peptides between 400 and 1000 m/z using a repeated cycle of 75 MS/MS scans with an 8 m/z-wide isolation width. Using a multiplexing scheme based on staggering the isolation window in alternating cycles of MS/MS scans, the precursor selectivity of the resulting data was 4 m/z after computational demultiplexing [21]. From the resulting data, fragment ion chromatograms were extracted for any peptide precursor between 400 and 1000 m/z. To determine which peptides were confidently detected in each acquisition, the software tool EncyclopeDIA [22] was used to query the data using a reference chromatogram library generated from a prior analysis of the tissue. Chromatogram-based relative quantitation was then performed for each peptide detected. All measurements were made relative to a common reference. In the case of plasma, a common reference was a pooled standard from 400 age-matched samples from subjects who were both healthy and spanning the phenotypes being studied, processed in each sample batch. Within each 96-well plate, one copy of the common reference standard was placed in each row for a total of 8 copies per plate [23]. Before quantification, it was confirmed that peptides were stable during the time scale of the data collection, had a suitable linear response, and had suitable intra- and inter-data precision.

**SomaScan.** SomaLogic, Inc. (Boulder, CO, USA) performed proteomic analysis of the plasma samples using the SomaScan platform, which quantifies the presence of 6995 human proteins (secreted proteins, extracellular domains, and intracellular proteins). These proteins are found in diverse biological groups such as receptors, kinases, cytokines, proteolytic enzymes and inhibitors, growth factors, protease inhibitors, hormones, and structural proteins [24]. Most of these proteins are involved in pathways related to signal transduction, stress response, immune processes, phosphorylation, proteolysis, cell adhesion, cell differentiation, and intracellular transport [25].

Proteins were measured using the Slow Off-Rate Modified Aptamer (SOMAmer)-based capture array, which uses chemically modified single-stranded DNA sequences capable of uniquely recognizing individual proteins via one or more binding sites. SOMAmers transform a protein signal to a nucleotide signal that is quantified using relative fluorescence on microarrays. Each SOMAmer array underwent validation for specificity, upper and lower limits of detection, and both intra- and inter-assay variability. Plasma dilutions (0.005%, 0.05%, and 20%) were applied to capture low-, medium-, and high-abundance proteins. Positive and negative controls were also positioned on the array to confirm if the experimental procedure was performed correctly.

After the hybridization step, the microarrays were washed and scanned using a laser scanner that excites the fluorescence of the fluorochrome used in the labeling step. The amount of the emitted signal is directly proportional to the amount of dye on the microarray. The scanner measured this quantity and created a digital image map of the position of each signal based on the location of its origin. Image analysis software generated a text file from this map describing the pixel intensity of the spot and the background. This information was processed, and a final value was generated that summarized the level of each detectable protein on the microarray.

Quality control analysis of the sample and SOMAmer levels involved SOMAmer control arrays on the microarray and calibration samples. Hybridization controls measured the sample-by-sample variation in hybridization, while the median signal over all SOMAmer arrays measured the technical variability. The SomaScan measures were reported as relative fluorescence units (RFU) in a summary ADAT file.

## Analysis and statistics

**Nordic collagen marker analysis.** Nordic collagen data turnover rates were calculated by performing ratios of the propeptide over the mature peptide (e.g., PRO-C1/ C1M). This ratio is an estimate of the formation/degradation of each collagen type and can be interpreted as a net gain of collagen/loss of collagen remodeling [18]. All comparisons were done using Student's T-test.

**SomaScan analysis.** A series of quality control and data cleaning steps was performed on the SomaScan proteomics data. All non-human proteins were excluded from the analyses. Samples that did not pass the default normalization acceptance criteria for all row scale factors were removed. All aptamers that did not pass the default quality control criteria across all plates/sets were filtered out. The data had no missing values, and the protein values were log-transformed. No further normalization was performed.

We identified differentially expressed aptamers using linear regression of each log-transformed protein as the outcome and disease status as the main predictor. We adjusted each analysis by age and sex. We analyzed all repeated measurements of each individual and employed generalized estimating equations to account for technical and biological variability using the gee package in R. Due to the high number of aptamers tested in the SomaScan assay (6995), we were at a statistical disadvantage when applying multiple corrections. Therefore, we calculated adjusted p-values using Benjamini-Hochberg methods. Our SomaScan analyses report only proteins with statistically significant differences from the control at an FDR-adjusted p-value<0.25.

**Mass spectrometry analysis.** For the MS protein data, all zeroes were replaced with NAs across samples, and protein values were log-transformed. The MS dataset was subset based on proteins that overlapped with the signature proteins associated with DD previously identified using the SomaScan proteomics data. We used a generalized estimation equation model to account for the repeated blood draws, adjusted by age and gender.

**Proteomic risk score calculations.** To prepare the data for the Diagnostic Proteomic Risk Score (DPRS) calculation, we first removed all highly correlated proteins (Pearson correlation>0.8). 3369 out of 6995 aptamers were highly correlated and were excluded from the DPRS calculation. The DPRS calculation contained the following two steps. First, we fit a logistic regression for each protein, with disease status (i.e., case or control) as the outcome variable, adjusted for age and gender. We used a generalized estimation equation model for repeated measurements of two independent blood draws. We included the same subject's repeated measurements and used the generalized estimation equations model with the gee R package [26] to estimate the regression coefficients. Next, the DPRS was computed using summary statistics from each regression model. Only the proteins with statistically significant differences from control at an FDR-adjusted p-value<0.25 were included in the score calculation. The DPRS was calculated by summing the beta estimated effect for all significant proteins, each multiplied by normalized protein abundance and divided by its standard error. Finally, we fit a logistic regression to examine the association of the computed DPRS with DD onset. We split the dataset into training (75%) and testing sets utilizing the caret R package for a leave-one-out cross-validation approach for model fitting and assessment (25%) sets [27]. We used R version 4.2.1 for all analyses.

   

**Pathway analysis.** STRING (12.0) is a web tool that provides and illustrates protein-protein interactions and networks [28]. STRING integrates evidence from public databases, including genomic context, high-throughput and biochemical interaction experiments, known protein complexes, computational predictions, and text mining from scientific literature. Given a list of gene names, STRING constructs protein-protein interaction networks in which each gene is a node, with edges representing predicted functional associations.

Networks were generated using the whole genome as the statistical background, and interaction sources included text mining, experiments, databases, co-expression, neighborhood, gene fusions, and co-occurrence. Interaction scores were computed by combining the probabilities across all evidence channels and correcting for the probability of observing an interaction by chance. All nodes and interactions were computed with a required medium confidence interaction score (0.400) and an FDR<=0.05.

## Results

### Plasma collagen metabolites in DD

Collagen synthesis and degradation processes release protein fragments specific to the process, collagen type, and enzyme involved. The ratio of synthesis to degradation-related fragments correlates with net accumulation or loss of that collagen type. We evaluated plasma markers of collagen synthesis and degradation using a competitive ELISA (Nordic Biosciences; Herlev, Denmark) and calculated collagen synthesis-to-degradation ratios [18].

We measured markers of types I, II, III, IV, V, VI, VII, and IX collagen metabolism in blood from a cohort of healthy controls (n = 14) and subjects with DD (n = 13). We found significantly lower levels of markers of Collagen I degradation (C1M; Fig 1A, p = 0.004), Collagen V synthesis (Pro-C5; Fig 1B, p = 0.046), and Collagen VII synthesis (Pro-C7; Fig 1C, p = 0.048) in the DD cohort compared to the control group. The remaining collagen biomarkers did not show significant differences between the disease and control groups (Table 1).

We calculated collagen synthesis-to-degradation ratios for collagen types I, II, III, IV, and V. The Collagen I synthesis-to-degradation ratio, Pro-C1/C1M, was significantly greater in the DD cohort than in the control group (Fig 1D, p = 0.024). The remaining ratios were not significant (Col II, III, IV, VI) or could not be calculated because the degradation marker was either unavailable (Col V, IX) or not detectable in both disease and control cohorts (Col VII).

### Differentially expressed blood proteins in DD

We used two orthogonal, unbiased proteomic methods: the aptamer-based SomaScan (SomaLogic; Boulder, CO) and liquid chromatography-tandem mass spectrometry (LC-MS/MS) (MacCoss Lab, University of Washington, WA, USA).

We used sixty-nine samples for SomaScan testing after quality control (36 DD and 33 controls). These represent 42 subjects, 27 of whom had second samples collected 6–14 months after the first (S1 Fig in S1 File). The SomaScan panel measures 6995 protein binding sites applying to 5444 proteins.

**Hypothesis-free approach.** The 6995 aptamer measurements identified 54 proteins with statistically significant differences between the DD and control groups, with 24 overexpressed and 30 underexpressed after multiple-comparison adjustment (S1 Table in S1 File). Fig 2 is a heatmap of these 54 protein expressions.

Using SomaScan results, we aimed to develop a hypothesis-free indicator of DD diagnosis, the Diagnostic Proteomic Risk Score (DPRS), to estimate an individual's disease risk based on plasma protein abundance.

**Hypothesis-free DPRS.** We developed a hypothesis-free DPRS estimation based on two of the 54 proteins. These proteins, OSCAR (overexpressed) and SPART (underexpressed), had a low relative correlation (Pearson's correlation coefficient = −0.19) (Table 2), suggesting minimal interaction. Using this hypothesis-free DPRS as a diagnostic measure for DD achieved 79% accuracy in the training set (AUC 0.8) and 76.5% in the test set (Fig 3B), indicating predictive value

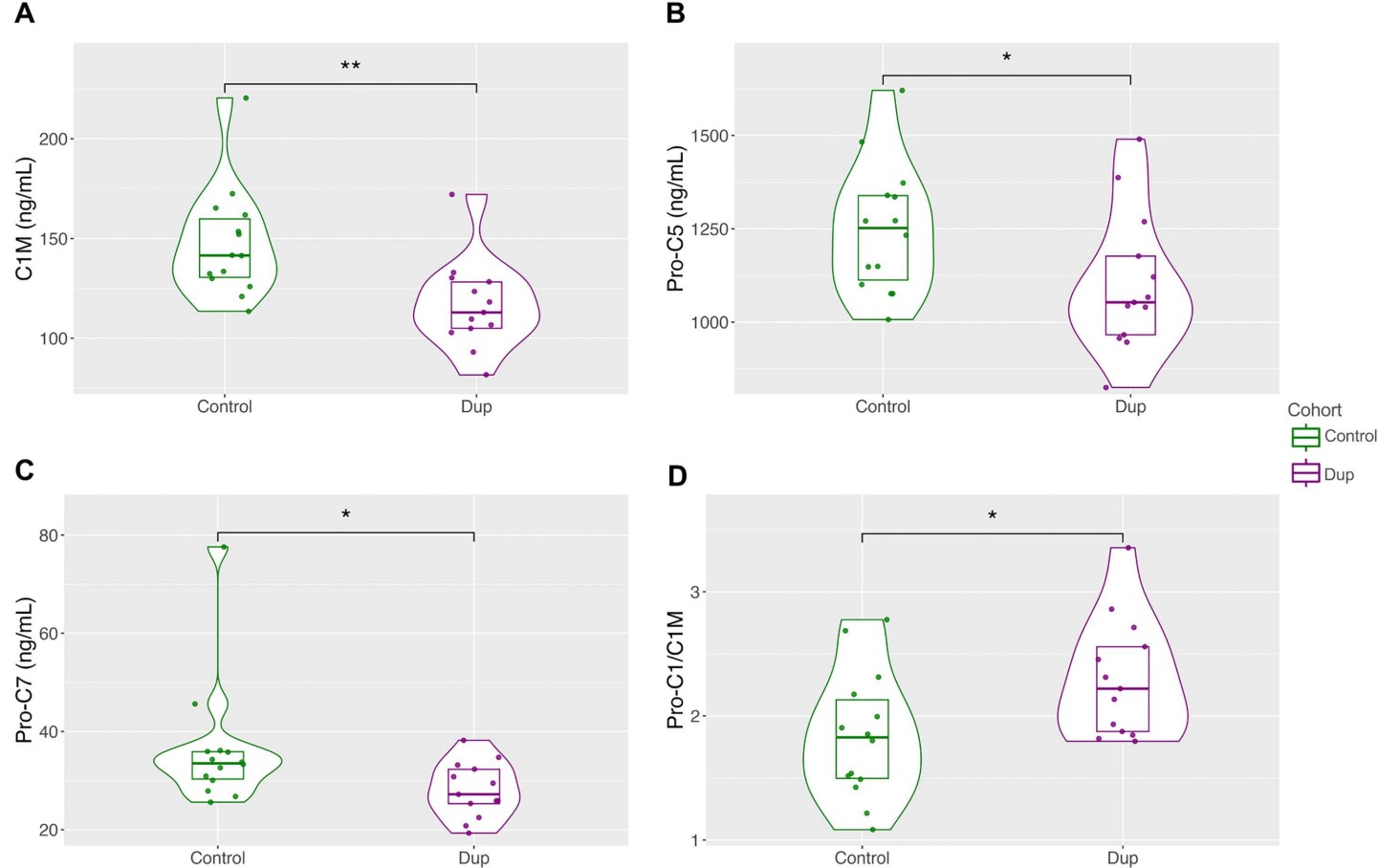

**Fig 1. Differential production of Dupuytren vs. control collagen metabolism markers. A)** Collagen I degradation marker C1M. **B)** Collagen I synthesis to degradation marker ratio Pro-C1/C1M. **C)** Collagen V synthesis marker Pro-C5. **D)** Collagen VII synthesis marker Pro-C7. * p-value <0.05, **p-value <0.005.

for discriminating between disease and control. Fig 3C shows a box plot of control and DD DPRS. The DPRS for controls and DD differed significantly by the Wilcoxon test (p-value = 2.5E-06).

**Hypothesis-based approach.** We also used a hypothesis-based approach, identifying 328 relevant proteins from a literature search and available on the SomaScan platform (S3 Table in S1 File). Twenty-three proteins in this subset showed statistically significant differences between DD and control, with nine overexpressed and 14 underexpressed after multiple-comparison adjustment (S4 Table in S1 File).

We constructed PPI networks and performed enrichment analysis of these 23 proteins using STRING. We found significantly enriched PPI in 21 proteins with multiple enriched categories (Fig 4A, p-value = 1.35E-07). We found more enrichment categories and greater significance in the hypothesis-based than the hypothesis-free analysis, which may reflect a higher-yield group, selection bias, or both. S5 Table in S1 File lists enriched gene categories, and S3 Fig in S1 File provides a visual cross-reference between functional categories and individual genes.

**Hypothesis-based DPRS.** We developed a hypothesis-based DPRS using 11 of 23 DE proteins identified in the hypothesis-based analysis. Six genes were underexpressed (*SMAD1, SH3BP2, LCN2, USP8, CSNK1G2,*

**Table 1. Collagen epitope markers and findings.**

| Assay | D/C | Description | Represents | p-value | SE | FC (D/C) |
|-------|-----|-------------|------------|---------|-----|----------|
| PRO_C1_HP | ns | Internal epitope in the N-terminal propeptide of type I collagen | Col I Synthesis | 0.9437 | C 10.99 D 11.89 | 0.93 |
| C1M_HP | Down** | MMP-generated fragment of the a1 chain of type I collagen | Col I Degradation | 0.0036 | C 5.25 D 4.30 | 0.73 |
| PRO-C2_HP | ns | N-terminal fragment of type IIB procollagen | Col II Synthesis | 0.7808 | C 1.45 D 3.11 | 1.01 |
| CALC2 | ND | C-terminal propeptide of type II collagen | Col II Synthesis | NA | NA | NA |
| C2M_HP | ns | MMP-generated fragment of type II collagen | Col II Degradation | 0.1607 | C 1.52 D 1.16 | 1.29 |
| PRO-C3_roHP | ns | N-terminal propeptide of type III collagen | Col III Synthesis | 0.9951 | C 1.04 D 1.24 | 0.93 |
| C3M_HP | ns | MMP-9 generated fragments of type III collagen | Col III Degradation | 0.1615 | C 16.96 D 13.61 | 0.86 |
| CTX-III_HP | ns | MMP-generated fragment of crosslinked type III collagen | Col III Degradation | 0.4887 | C 40.88 D 53.26 | 0.91 |
| PRO-C4_HP | ns | Internal epitope in the 7S domain of type IV collagen | Col IV Synthesis | 0.0739 | C 195.98 D 183.12 | 0.84 |
| C4M_HP | ns | MMP-generated fragment of the a1 chain of type IV collagen | Col IV Degradation | 0.1014 | C 8.89 D 7.42 | 0.84 |
| PRO-C5_HP | Down* | C-terminal propeptide of type V collagen | Col V Synthesis | 0.0458 | C 33.17 D 35.88 | 0.82 |
| PRO-C6_roHP | ns | C-terminal propeptide of type VI collagen alpha-3 chain | Col VI Synthesis | 0.7222 | C 0.27 D 0.27 | 0.92 |
| C6M_HP | ns | MMP2-generated fragment of type VI collagen | Col VI Degradation | 0.1097 | C 22.31 D 23.41 | 0.83 |
| PRO_C7 HP | Down* | BMP-I mediated neo-epitope in the NC-2 propeptide of type VII collagen | Col VII Synthesis | 0.0480 | C 2.49 D 1.08 | 0.72 |
| C7M | ND | Neo-epitope of MMP13 degradation of type VII collagen | Col VII Degradation | NA | NA | NA |
| PRO-C9 | ns | C-terminus of NC-I domain of the a1 chain of type IX collagen | Col IX Synthesis | 0.9658 | C 0.80 D 0.65 | 0.92 |

Table 1 synthesis and degradation markers of Collagen I, II, III, IV, V, VII, and IX. Compared to controls, there were significant reductions in markers of Collagen I degradation, Collagen V synthesis, and Collagen VII synthesis. D/C: ratio of Dupuytren to control values. FC: fold change. **: <0.005, *: <0.05. ns: not significant. ND: undetectable in both disease and control cohorts.

SERPINH1) and five overexpressed (*CSMD1, POSTN, ACAN, AOC3, DDR2*). Six proteins were present in hypothesis-based and hypothesis-free DPRS analyses (Table 3). Our hypothesis-based DPRS achieved a training accuracy of 78.85% (AUC 0.9) and a test accuracy of 70.59% in distinguishing Dupuytren from control cohorts (Fig 4B). Fig 4C shows that DPRS distributions of Control and DD groups differ significantly (Wilcoxon test, p-value = 2.2E-09).

**Evaluation of DRPS for prognosis.** We explored the potential of each DPRS to profile DD prognosis. We used age at the time of the first DC procedure as an index of biological severity. We segregated our DD samples into subjects younger (11 specimens) or older (16 specimens) than age 50 at the time of their first DC procedure. We excluded from this analysis subjects younger than 50 at the time of their blood draw. A one-way ANOVA revealed a statistically significant difference in the means of the hypothesis-based DPRS across groups with different rates of pretreatment contracture progression (S4 Fig in S1 File; p-value = 0.0018). Conversely, we found no associations in the DPRS generated through the hypothesis-free approach.

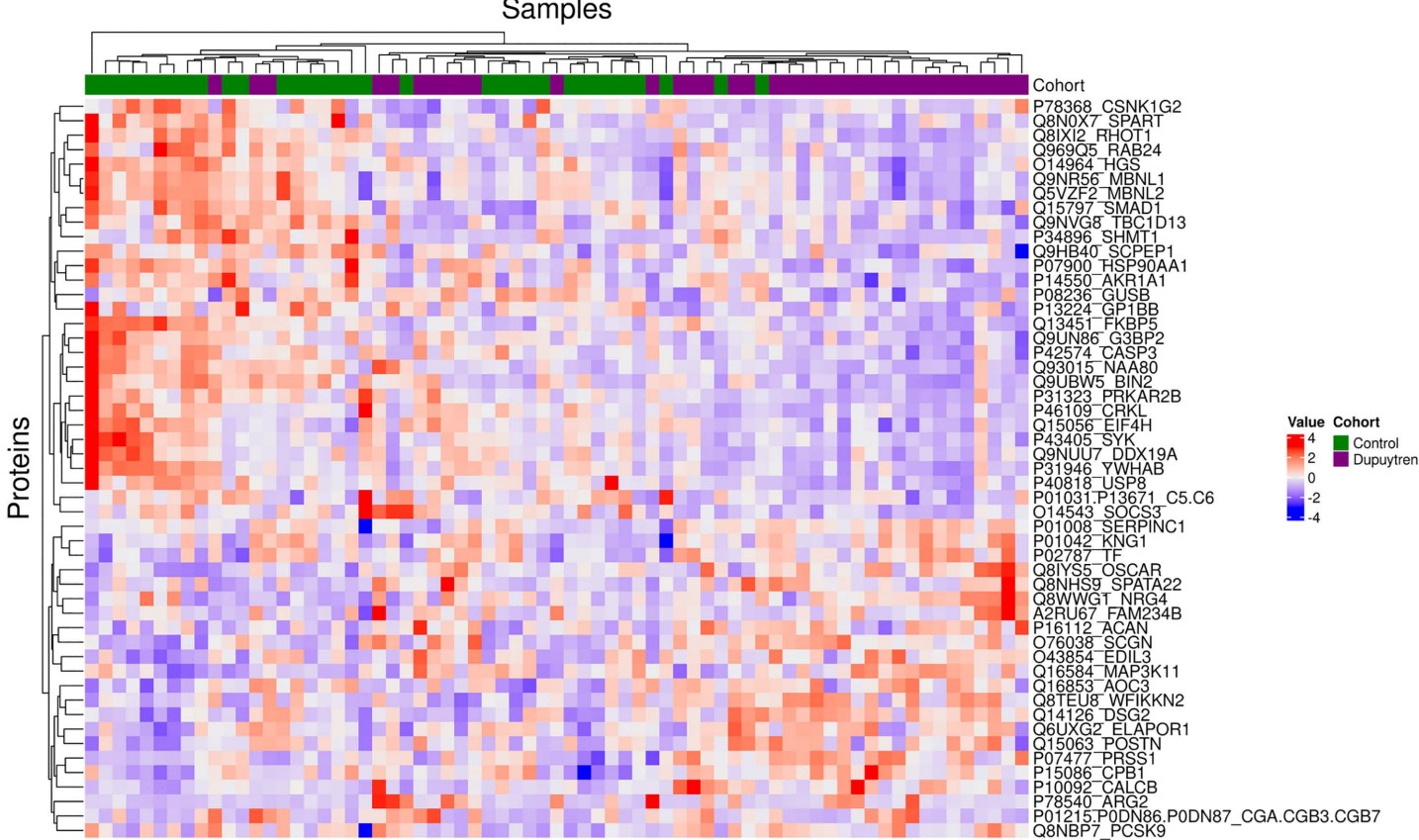

**Fig 2. Heatmap of the 54 significant proteins in the hypothesis-free analysis.** This heatmap shows natural log-transformed and standardized relative fluorescence units (RFU) for proteins (rows) across samples (columns). Red colors represent higher-than-average RFU values, and blue represents lower-than-average RFU values. Rows and/or columns are organized using hierarchical clustering, and cohorts are indicated by the top bar [purple: Dupuytren; green: Control]. We performed pathway analysis of these 54 proteins to identify network relationships using STRING [28]. We found significant protein-protein interactions (PPI) in 29, with multiple enriched categories (Fig 3A, p-value = 4.55E-04). S2 Table in S1 File summarizes categories of enriched gene groups, and S2 Fig in S1 File provides a visual cross-reference of functional categories and individual genes. The dominant categories were coagulation and complement cascades, the extracellular matrix, enzyme inhibition, cell membrane structures, phosphorylation, and signaling pathways.

**Table 2. Hypothesis-free diagnostic proteomic risk score.**

| Protein | Eff | SE | p-value | FDR |
|---------|-----|-----|---------|-----|
| Spartin | −11.62125 | 2.97479 | 0.000094 | 0.18 |
| Oscar | 13.99037 | 3.596067 | 0.0001 | 0.18 |

Table 2 proteins **of the hypothesis-free DPRS**. Eff = log (odds ratio comparing Dup to control) for a change of 1 in the log (protein data). SE: Standard error. FDR: False Discovery Rate (Adjusted p-value).

**Mass spectrometry.** The LC-MS dataset contained 48 samples: 27 DD and 21 controls. This assay identified 561 peptides. No peptides had statistically significant differences in abundance across groups (FDR > 0.25).

**Mass-spectrometry-SomaScan concordance.** To further validate our SomaScan findings, we evaluated the MS results for concordance. Of the 54 significant proteins identified in the SomaScan analysis, six were also detected in the

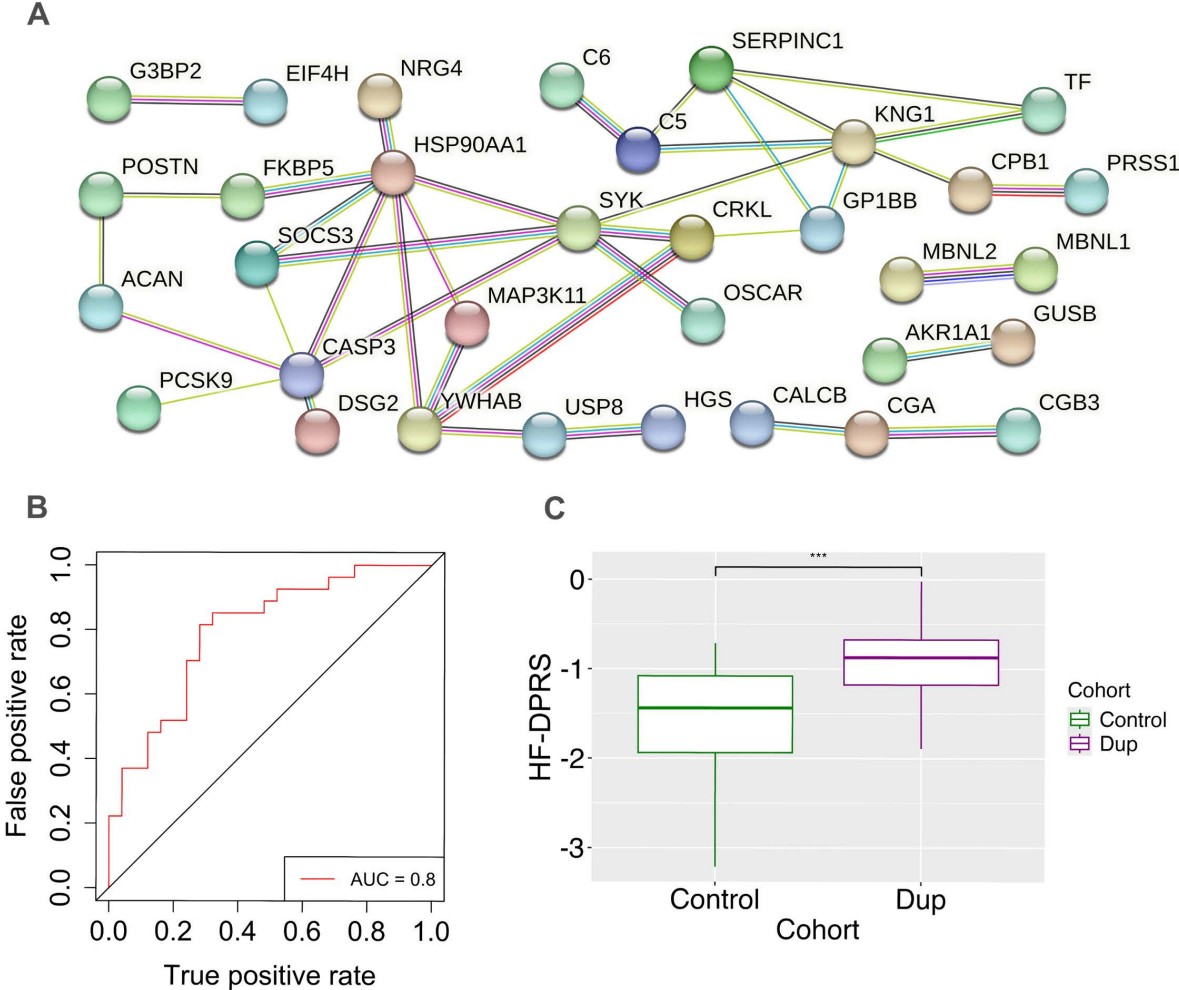

**Fig 3. Hypothesis-free DPRS and protein-protein interactions. A)** Network expression of 54 differentially expressed proteins. Of the 54 proteins, 30 had strong pathway connections in the network: ACAN, AKR1A1, C5, C6, CALCB, CASP3, CGA, CGB3, CPB1, CRKL, DSG2, EIF4H, FKBP5, G3BP2, GP1BB, GUSB, HGS, HSP90AA1, KNG1, MAP3K11, MBNL1, MBNL2, NRG4, OSCAR, PCSK9, POSTN, PRSS1, SERPINC1, SOCS3, SYK, TF, USP8, and YWHAB. **B)** The Hypothesis-free Diagnostic Proteomic Risk Score has an AUC-ROC value of 0.8. **C)** Hypothesis-free Diagnostic Proteomic Risk Scores (HF-DPRS) of the control vs Dupuytren cohorts. p-value = 5.28E-06.

Mass Spectrometry findings: AOC3, POSTN, SERPINC1, TF, PCSK9, and KNG1. We ran the same analyses on these six proteins using Mass Spectrometry data. AOC3, POSTN, and SERPINC1 had adjusted p-values (FDR) <0.25 in both Mass Spec and SomaScan analyses (Table 4).

## Discussion

This study aimed to identify candidate biomarkers that correlate with DD diagnosis as a first step toward a diagnostic DD blood test.

Dynamic biomarker panels are essential for developing preventive therapeutics for chronic diseases such as DD. The relevance of this study is that DD lacks such biomarkers. Blood is the most promising unexplored source of proteomic biomarkers for trend analysis.

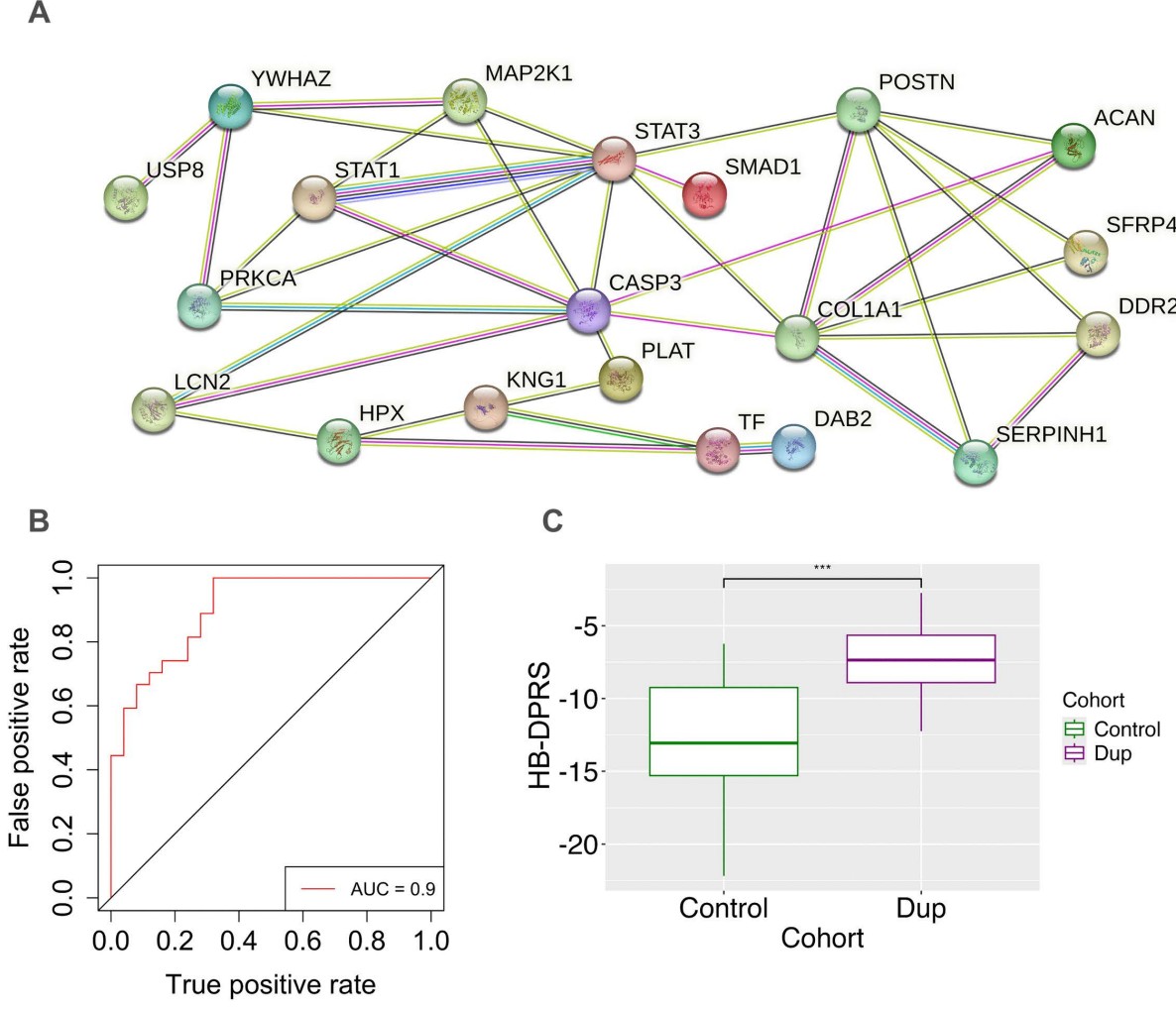

**Fig 4. Hypothesis-based DPRS and protein-protein interactions. A)** Network expression of 23 differentially expressed proteins. Of the 23 proteins, 20 had strong pathway connections: DAB2, SMAD1, USP8, PLAT, SFRP4, ACAN, DDR2, HPX, KNG1, LCN2, SERPINH1, TF, MAP2K1, PRKCA, STAT1, YWHAZ, POSTN, COL1A1, CASP3, and STAT3. **B)** The Hypothesis-based Diagnostic Proteomic Risk Score has an AUC-ROC value of 0.9. AUC-ROC (Area Under the Curve – Receiver Operating Characteristics) is an index of predictive accuracy. **C)** Hypothesis-free Diagnostic Proteomic Risk Scores (HF-DPRS) of the control vs Dupuytren cohorts. (p-value = 7.79E-09).

There are no broad surveys of circulating proteins in DD. In addition, blood findings may not parallel tissue findings because of rapid degradation, intracellular or membrane-bound locations, post-translational variants, protein binding, and other mechanisms. Because of this, new and diverse discovery research is necessary to profile DD-associated blood protein biomarkers.

There are two common strategies for biomarker discovery: hypothesis-free and hypothesis-based. The hypothesis-free approach (also called unbiased or non-hypothesis-based) screens many markers and is the standard in systems biology and discovery research. The hypothesis-based strategy evaluates selected markers based on published information and is the most common model of DD research. We used both methods in this study.

**Table 3. Hypothesis-based diagnostic proteomic risk score.**

| Protein | Eff | SE | p-value | FDR |
|---|---|---|---|---|
| SMAD1* | −6.3649 | 1.877804 | 0.0007 | 0.12 |
| CSMD1 | 5.397535 | 1.907423 | 0.00466 | 0.18 |
| SH3BP2 | −4.59151 | 1.582341 | 0.00371 | 0.18 |
| POSTN* | 4.305245 | 1.502627 | 0.00417 | 0.18 |
| ACAN* | 5.589134 | 1.75746 | 0.00147 | 0.12 |
| LCN2 | −4.73895 | 1.734879 | 0.0063 | 0.19 |
| AOC3* | 7.050175 | 2.62144 | 0.00716 | 0.2 |
| DDR2 | 1.63418 | 0.595212 | 0.00604 | 0.19 |
| USP8* | −3.30095 | 1.041269 | 0.00152 | 0.12 |
| CSNK1G2* | −10.3331 | 3.593633 | 0.00404 | 0.18 |
| SERPINH1 | −2.6813 | 0.799538 | 0.0008 | 0.12 |

Table 3 proteins of the hypothesis-based DPRS. Eff = log (odds ratio comparing Dup to control) for a change of 1 in the log (protein data). SE: Standard error. FDR: False Discovery Rate (Adjusted p-value). * Six also had significant differential expression in the 54 proteins identified in the hypothesis-free analysis.

**Table 4. Mass spectrometry (MS) – SomaScan (SS) concordance.**

| Protein | Est MS | p-val MS | FDR MS | Est SS | p-val SS | FDR SS |
|---|---|---|---|---|---|---|
| AOC3 | 0.214854 | 0.076413 | 0.232677 | 0.116624 | 0.000845 | 0.217972 |
| POSTN | 0.410212 | 0.098842 | 0.232677 | 0.182892 | 0.000312 | 0.211479 |
| SERPINC1 | 0.071331 | 0.116338 | 0.232677 | 0.091474 | 0.001500 | 0.225251 |
| TF | 0.035200 | 0.567393 | 0.829196 | 0.076715 | 0.000025 | 0.172072 |
| PCSK9 | 0.165811 | 0.755318 | 0.829196 | 0.185675 | 0.001371 | 0.222267 |
| KNG1 | 0.010065 | 0.829196 | 0.829196 | 0.065561 | 0.000490 | 0.217972 |

Table 4 proteins identified in both MS and SS analyses. Six proteins identified by MS were also in the 54 proteins found in the hypothesis-free SS analysis: AOC3, POSTN, SERPINC1, TF, PCSK9, and KNG1. Of these, AOC3, POSTN, and SERPINC1 had False Discovery Rate (FDR, Adjusted p-value) <0.25 in both Mass Spec and SomaScan analyses. Est MS: Natural log fold change in protein abundance between Dupuytren disease cases and controls, measured by Mass Spectrometry, estimated using a generalized estimating equation (GEE) model adjusted for age and gender. p-val MS: P-value for the Est MS, calculated from the robust standard error of the GEE model. FDR MS: False discovery rate (FDR)–adjusted p-value for Mass Spectrometry results, computed using the Benjamini–Hochberg procedure. Est SS: Natural log fold change in protein abundance between Dupuytren disease cases and controls, measured by SomaScan, estimated using a generalized estimating equation (GEE) model adjusted for age and gender. p-val SS: P-value for the Est SS, calculated from the robust standard error of the GEE model. FDR SS: False discovery rate (FDR)–adjusted p-value for SomaScan results, computed using the Benjamini–Hochberg procedure.

## Plasma collagen metabolites in DD

Collagen accumulation is the hallmark of DD. DD-affected tissues have unique proportions of collagen types I [29], III [30], IV [16], and V [29]. The most prominent tissue changes in the progression from preclinical to end-stage disease are increases in total Collagen I and the proportion of Collagen III [29].

Our hypothesis-based approach compared markers of collagen synthesis and degradation between the DD and control groups. Collagen turnover releases circulating peptides unique to the synthesis and degradation of each collagen type. In the absence of disease, synthesis and degradation markers are tightly coupled. Plasma synthesis-to-degradation marker ratios can indicate net collagen gain, loss, or homeostasis [18]. We used ELISA to measure plasma levels of 16 collagen metabolism markers across 8 collagen types. We summarize our findings in Table 1.

The DD cohort had normal Pro-C1 levels, a marker of Collagen I synthesis, but showed a statistically significant decrease in C1M, a marker of Collagen I degradation (Fig 1A), resulting in an elevated synthesis-to-degradation marker

ratio in DD compared to controls (Fig 1D). These findings suggest DD collagen accumulation follows impaired Collagen I degradation rather than excess collagen synthesis. Both degradation-resistant collagen variants and reduced collagenase activity may contribute to reduced collagen degradation.

Dupuytren collagen has two features that increase resistance to enzymatic degradation. Typical Collagen I fibrils are heterotrimers of two Collagen I α1 strands and one Collagen I α2 strand. DD-affected tissues have an increased proportion of α1(I) homotrimeric Collagen I [31]. DD collagen also has more crosslinks stabilized by hydroxylation and glycosylation [32]. Both homotrimer Collagen I [33] and glycosylation-stabilized crosslinks [34] increase resistance to enzymatic cleavage.

DD-related dysregulation of collagenases and their inhibitors impairs collagen degradation by several mechanisms. TIMP1 (Tissue Inhibitor of Metalloproteinase 1) inhibits multiple MMPs and is elevated in the circulation of DD patients [35]. MMP1, MMP8, MMP13, and MMP14 gene polymorphisms are associated with Dupuytren disease [36,37], including MMP14 partial loss-of-function variants [37]. MMP14 is necessary for MMP2 activation, and both enzymes facilitate myofibroblast contraction [38].

Two clinical findings support a relationship between impaired collagen degradation and DD. The first is the rapid onset of DD during treatment with a metalloproteinase inhibitor [39]. The second is abnormally slow matrix remodeling in DD. Enzymatic cleavage of collagen fibrils is necessary for matrix shortening [40], and impaired cleavage would slow this process. Slow matrix remodeling could explain why DD-related tissue shortening progresses more slowly than wound healing despite similar biology, and why DD surgery is associated with an increased risk of delayed wound healing [41]. The role of impaired collagen degradation in pathogenesis and intervention is unclear, and simply normalizing collagen degradation might have the undesirable effect of accelerating tissue contraction.

The hallmark of fibrotic diseases is the accumulation of collagen. Increased collagen synthesis contributes to collagen accumulation in pulmonary fibrosis [42], liver fibrosis [43], primary biliary cirrhosis [44], renal fibrosis [45], and cardiovascular disease [46]. Our finding of normal levels of circulating Collagen I synthesis biomarkers in DD suggests that the mechanism of collagen accumulation in DD is not due to the increased collagen synthesis associated with these diseases. Although often used interchangeably with increased collagen synthesis, neither collagen accumulation nor upregulated collagen gene expression is, in itself, evidence of increased collagen synthesis. Cell culture studies show increased COL1A1 and COL1A2 mRNA expression in DD-derived fibroblasts [47,48], but these RNA findings are not associated with increased collagen I protein production [48]. Our findings of normal Collagen I synthesis and impaired degradation inform future drug development research beyond the traditional focus of reducing collagen synthesis [49] to normalizing collagen degradation.

Collagen II markers were normal. We evaluated Collagen II markers because of the correlation between DD and psoriasis [50] and the finding of abnormal Collagen II in psoriasis [51]. We found no significant differences between groups in the Collagen II synthesis marker PRO-C2_HP, the degradation marker C2M_HP, or their ratios. The CALC2 synthesis marker was undetectable in both cohorts, preventing the calculation of CALC2 synthesis-to-degradation ratios.

Collagen III synthesis markers were normal, consistent with a prior report of normal circulating Collagen III synthesis marker amino-terminal propeptide of type III procollagen (PIIINP) in DD subjects [52]. Given the accumulation of Collagen III in DD, we expected to find elevated Collagen III synthesis-to-degradation marker ratios. Instead, we found normal Collagen III synthesis-to-degradation ratios. This finding may reflect that other proteases also degrade Collagen III, and we did not measure all potential collagen degradation fragments. To investigate further, we assayed a different Col III degradation marker, CTX-III_HP, on the remaining aliquots. Only 17 (10 DD, 7 Controls) of the initial 27 (13 DD, 14 Controls) leftover aliquots had sufficient residual volumes for this assay. Although this small subgroup's CTX-III violin plot appearance suggested reduced degradation in DD compared to controls (S5 Fig in S1 File), this difference was not statistically significant.

We found significantly lower Pro-C5 levels, a marker of Collagen V synthesis, in the DD cohort than in controls (Fig 1B). This finding is unexplained. An increased percentage of Collagen V is reported in DD tissues [29], suggesting a high Collagen V synthesis-to-degradation ratio. However, we could not calculate this ratio because a Collagen V degradation marker was unavailable.

The DD cohort also had significantly lower levels of the Collagen VII synthesis marker Pro-C7 than controls (Fig 1C). We could not calculate Collagen VII synthesis-to-degradation ratios because Collagen VII degradation markers were undetectable in both cohorts. Although there are no reports of Collagen VII in DD, its role in matrix mechanobiology and elevated tissue levels in systemic sclerosis [53] make it a potential marker for collagen metabolism abnormalities in DD.

We found no significant differences in the remaining collagen markers. We measured Collagen IV markers because Collagen IV is elevated in DD tissues [16]. We assayed Collagen VI markers and ratios because prior pathway analysis suggests a role for Collagen VI in DD and other fibrotic diseases [54]. We selected Collagen IX markers based on unpublished data suggesting a role in DD. None of these markers or ratios showed significant differences between disease and control cohorts.

## Plasma proteins and interactions in DD

In addition to the Nordic collagen fragment analysis, we analyzed whole proteins using the SomaScan aptamer-based platform. The SomaScan panel measures 6995 protein-binding sites associated with 5444 proteins. S6 Table in S1 File lists genes and gene aliases of proteins mentioned in this manuscript.

Our SomaScan hypothesis-free analysis identified 54 distinct proteins with plasma levels that were significantly different from those of controls. Twenty-four proteins showed increased levels, and 30 showed lower DD levels than controls (S1 Table in S1 File). Our findings are consistent with existing literature.

The body of DD research publications mentions only seven of these 54 differentially expressed proteins: Aggrecan (*ACAN*), Amine Oxidase Copper Containing 3 (*AOC3*), Caspase-3 (*CASP3*), Periostin (*POSTN*), SMAD Family Member 1 (*SMAD1*), Spartin (*SPART*), and Ubiquitin carboxyl-terminal hydrolase 8 (*USP8*). Our differential expression findings concur with reports of *ACAN* [55], *AOC3* [13], and *POSTN* [56] overexpression in tissue and *SMAD1* [57], *SPART* [58], and *USP8* [58] underexpression in cell culture. CASP3 levels were low in our findings, but are upregulated in a transcriptomic profiling report [55]. This finding is unexplained. *ACAN* is also near the variant rs6496519, which is associated with DD [59]. S1 Table in S1 File summarizes the relevance of these 54 proteins and their parent genes to DD.

We constructed PPI networks and enrichment analysis of these 54 plasma proteins, revealing significantly enriched protein-protein interactions in 32, with multiple enriched protein groups and categories (S2 Fig, S2 Table in S1 File). These categories align with known Dupuytren cellular biology. Abnormalities in the KEGG category has04610 (complement and coagulation cascades) were consistent with DD pathway meta-analysis findings [60]. Complement fragments induce adhesion molecules that act on endothelial cells, consistent with DD-related perivascular clusters of immune cells and small vessel occlusion [13]. Complement fragments also activate mast cells that release TNF, a known DD pathway component [61], and histamine, a driver of itching, which is common in early-stage DD [62]. Cell membrane and extracellular matrix signaling receptor binding of growth factors, cytokines, and other signaling factors play pivotal roles in DD pathobiology [14]. Protein phosphorylation processes are consistent with increased phosphorylated STAT1, STAT3, SMAD2, SMAD3, and ERK1/2 in DD tissues [57,63,64]. The enriched Reactome HSA-162582 (Signal Transduction) pathway is noteworthy, as it includes signaling by TGF-Beta, WNT, and NOTCH, all of which are prominent in DD biology [65].

We based our hypothesis-free DPRS on high levels of Osteoclast-associated immunoglobulin-like receptor (*OSCAR*) and low levels of Spartin (*SPART*) (Table 2). Our finding of elevated OSCAR levels is interesting because Collagen I is an OSCAR ligand. OSCAR also stimulates T-cells to release TNF [66], a component of DD biology [61]. Our finding of decreased *SPART* levels is consistent with low *SPART* expression in DD fibroblasts [58]. SPART is also associated

with Epidermal Growth Factor Receptor (EGFR) degradation and transport to the cell membrane [67], consistent with increased ratios of intracellular to cell membrane EGFR levels observed in DD [68].

In addition to the hypothesis-free analysis, we conducted a hypothesis-based analysis using a subset of SomaScan proteins selected from an independent literature review (S3 Table in S1 File). We identified 23 proteins in this subset with significantly different plasma levels compared to controls. Nine proteins had higher levels in DD than in controls, and fourteen had lower levels. Nine proteins were present in both hypothesis-free and hypothesis-based groups. Our findings are consistent with existing literature. S4 Table in S1 File summarizes the potential relationships among these proteins, their parent genes, and DD.

Similar to our hypothesis-free analysis, we performed a STRING PPI analysis of the hypothesis-based findings to identify network relationships of the 23 proteins. We found significantly enriched protein-protein interactions in 21 with multiple enriched categories (p-value = 1.35E-07, S5 Table in S1 File). This analysis implicated additional enrichment categories beyond those from the hypothesis-free analysis. Myofibroblast differentiation, proliferation, motility, WNT signaling, and interleukin pathways are unsurprising, given their known roles in DD biology [10,69]. Cancer pathway associations are also unsurprising, given the correlations between DC and carcinomas and sarcomas [70,71]. Infection-related pathways were unexpected and were exclusively associated with viral and parasitic infection categories. This finding might be due to a hidden variable of tissue repair biology, which overlaps interleukin-driven parasitic immune defense biology and interleukin-related DD biology.

This hypothesis-based analysis generated data for a second Diagnostic Proteomic Risk Score (DPRS) calculated from 11 proteins (Table 3). Fig 4A shows the AUC-ROC of this panel (AUC = 0.9), and Fig 4B shows the DPRS based on this panel, distinguishing DD from control cohorts.

We evaluated the potential of each DPRS for staging. Recurrence after a procedure is the benchmark of Dupuytren's biological severity. Traditionally, surgeons use clinical factors such as early age of onset, affected family members, and disease locations other than the palms to predict post-procedure recurrence. However, the study's inclusion criteria included these factors, so we could not use them to subset our Dupuytren cohort. Instead, we used age at the time of the first DC procedure. Recurrence is more common in patients who are younger at their first DC procedure [72–76]. We evaluated each DPRS to distinguish Dupuytren subjects who had their first corrective procedure before age 50 from those who had their first correction at age 50 or older. The hypothesis-based DPRS distinguished DD subjects who underwent their first DC procedure before age 50 from those who underwent their first DC procedure after age 50 (p = 0.0018), suggesting potential use as a staging and predictive tool. An unexpected and unexplained finding is that DPRS scores were higher in DD than in controls, but hypothesis-based DPRS scores were higher in DD patients who had their first procedure after age 50 than in those who were younger at the time of their first corrective procedure.

Although Mass Spectrometry did not reveal statistically significant differences in abundance across cohorts, there was statistically significant concordance between Mass Spec and SomaScan findings. The Mass Spec data contained 6 of the 54 proteins with statistically significant differences between disease and control cohorts using the hypothesis-free SomaScan data analysis: Amine Oxidase Copper Containing 3 *AOC3*, Periostin *POSTN*, Serpin Family C Member 1 *SERPINC1*, Transferrin *TF*, Proprotein Convertase Subtilisin/Kexin Type 9 *PCSK9*, and Kininogen 1 *KNG1*. Table 4 compares the results of the SomaScan and Mass Spectrometry data analysis. A pathway analysis of these six proteins identified two highly significant Reactome categories (S7 Table in S1 File). In addition to protein phosphorylation noted above, Reactome HSA-381426 involving Insulin-like Growth Factor *IGF* and Insulin-like Growth Factor Binding Proteins *IGFBP*s is relevant because of the role of IGF-II and IGFBP-6 in regulating Dupuytren fibroblast proliferation and contractility [77].

This study has limitations. Statistical disadvantages included the small sample sizes across all assays, and the large number of aptamers in the SomaScan assay. Although our findings align with current research on Dupuytren proteomics, there are differences. Elevated serum TIMP1 [35] and TNF [69] have been reported in DD, but we found no statistically

significant DE between disease and control groups. This discrepancy may reflect that SomaScan assays target specific protein conformations, whereas ELISA measures all conformations.

## Conclusions

We used ELISA, aptamer, and mass spectrometry assays to identify differential plasma proteomic signatures between DD and control cohorts. We found significant DD-related proteomic abnormalities in both circulating collagen metabolism peptide fragments and whole proteins. This study is the first to provide evidence suggesting net collagen accumulation in DD results from normal collagen synthesis and impaired collagen degradation, challenging the model of excess collagen synthesis and distinguishing DD from other organ fibroses. We independently analyzed proteins from the same cohorts. Our hypothesis-free analysis of 5444 proteins revealed 54 with statistically significant differences between the DD and control cohorts. Our independent hypothesis-based analysis of 328 proteins selected from literature review identified 23 with statistically significant differences. We developed two Dupuytren Diagnostic Proteomic Risk Scores, one based on hypothesis-free and the other based on hypothesis-driven analyses. We used pathway analyses to identify enrichments in the DD protein-protein network, confirming known DD interactions and revealing previously unreported DD pathway categories. Finally, we showed our hypothesis-based DPRS correlated with the subject's age at the time of the first DD procedure, an index of biological severity. These findings suggest that a validated, hypothesis-based DPRS could guide management and the development of DD therapeutics.

We recommend a larger study to validate our findings on the metabolites of Collagen I, III, V, and VII. We also recommend a validating study of 68 unique proteins identified in the protein aptamer analysis (54 in the hypothesis-free analysis and 23 in the hypothesis-based analysis, with 9 in both groups). Testing only the subset of the 68 most pertinent proteins would increase statistical power. Individual ELISA assays would be cost-prohibitive. An alternative approach would be to develop a custom multiplex panel for the 13 proteins in our two DPRS [14] to validate our DPRS and to assess the correlation with our hypothesis-based DPRS on age at the first DD procedure. Our findings strongly support the need to fund and conduct more extensive validating studies of circulating collagen metabolites and proteins in DD. This biomarker research is fundamental to developing DD diagnostic tests, staging DD, and informing the development of therapeutic drugs for DD.

## Supporting information

**S1 Fig. Distribution of samples to laboratories.** Forty-five subjects provided initial samples. Seventeen of these subjects had second samples drawn at least 6 months after the first. Mass spec lab 1 performed analysis on the initial 45 samples. Mass spec lab 2 and collagen metabolite analyses were performed on redraw specimens. Aptamer-based analysis was performed on both initial and redraw specimens. Three of the original forty-five samples for Aptamer analysis failed quality control, leaving 23 DD and 19 Controls available for analysis.
(TIF)

**S2 Fig. Protein-protein network enrichment categories in the hypothesis-free analysis.** Thirty-two differentially expressed proteins showed significant network enrichment (p.adj < 0.05) in the following biological categories. C&C: coagulation and complement cascades, ECM: extracellular matrix, ENZ: Enzyme inhibition, MBR: cell membrane structures, PPL: phosphorylation, SGN: other signaling pathways.
(TIF)

**S3 Fig. Protein-protein network enrichment categories in the hypothesis-based analysis.** Twenty-one differentially expressed proteins showed significant network enrichment (p.adj < 0.05) in the following biological categories. CAN: Cancer, C&C: coagulation and complement cascades, ECM: extracellular matrix, INL: interleukin interactions, INF: infection, MBR: cell membrane structures, MFB: myofibroblast differentiation, proliferation, and motility, PPL: phosphorylation, SGN:

other signaling pathways, WNT: WNT pathways. Hypothesis-based selection bias from a preselected list of candidate proteins may influence the enrichment results.
(TIF)

**S4 Fig. Hypothesis-based DPRS correlates with age at the time of the first Dupuytren procedure.** This 11-protein Hypothesis-Based Dupuytren Proteomic Risk Score (HB-DPRS) distinguished DD subjects with different disease progression rates based on whether the subject was younger than 50 vs. 50 or older during their first corrective procedure (p = 0.0018).
(TIF)

**S5 Fig. Differential expression of Dupuytren vs. control Collagen III degradation marker CTX-III.** Although these violin plots visually suggest reduced Collagen III degradation marker CTX-III values in DD compared to controls, this difference was not significant (p-value = 0.4887). These data were from 17 of the 27 samples used for the other collagen metabolism markers, due to insufficient volumes in 10 samples. Because small cohorts magnify the effects of outliers, we recommend repeating these assays on larger cohorts.
(TIF)

**S1 Table. 54 Differentially expressed genes in the SomaScan Hypothesis-free analysis.** Of the 6995 aptamers in the Hypothesis-free analysis, 54 proteins showed statistically significant differences between the DD and control groups, with 24 overexpressed and 30 underexpressed after adjusting for multiple comparisons. Nine of these proteins were identified in both hypothesis-free and hypothesis-based analyses, as indicated by *. Plain italics indicate an indirect DD relationship, if any, to the protein; bold italics indicate a direct relationship between published DD findings and the protein. In the Notes column, "Referred to" means the referenced publication referred to the gene by an alias name. Exp: DD expression compared to controls. p-val: p-value of expression difference. FDR: False discovery rate (adjusted p-value. Nodes: number of pathway analysis connections (nodes) of this protein to others in this group with a medium confidence interaction score (0.400) and an FDR<=0.05. Notes: potential relationship to DD biology. Cat: categories of possible relationships to DD: 1. Apoptosis and senescence (3 genes); 2. Bioinformatics (9 genes); 3. Clinical and demographic (6 genes); 4. Extracellular matrix (6 genes); 5. Fibroblast and myofibroblast cytoskeleton, membrane, and motility (10 genes); 6. Fibroblast and myofibroblast differentiation and transcriptome (7 genes); 7. Vascular and perivascular (9 genes); Unclear relationship (16 genes).
(DOCX)

**S2 Table. Functionally enriched pathways derived from the 54 differentially expressed proteins in the hypothesis-free analysis.** ID: Enriched category name. STR: Strength of protein-protein interaction (PPI). FDR: False Discovery Rate (Adjusted p-value). Overall PPI enrichment p-value: 0.000455. Figure 4 summarizes enriched pathways for individual genes.
(DOCX)

**S3 Table. 328 protein-coding genes in the SomaLogic hypothesis-based analysis.** We combined three literature-based search strategies to identify candidate Dupuytren-related genes for hypothesis-based analytics. We first collected 2547 full-text publications from 1980 to 2023 that included at least three instances of the search terms "Dupuytren*" and/ or "palmar fibroma*". In the first strategy, we searched each publication for 4527 potential Dupuytren-related genes and their 33907 gene name aliases. This search identified 326 unique proteins or parent genes mentioned at least three times in at least one publication. The second method identified 117 selected collagen metabolism-related proteins or their parent genes that appeared at least once in at least one of these publications. The third method identified potentially DD-related single-nucleotide polymorphisms (SNPs) reported in these publications and compiled a list of 369 genes adjacent to these SNPs. We merged these three lists, resulting in 546 unique protein-coding genes, 328 of which matched proteins on the

SomaScan panel, which contains 6995 protein-binding sites. Proteins expressed by these 328 genes were the targets of our hypothesis-based analysis.
(DOCX)

**S4 Table. 23 Differentially expressed genes in the Hypothesis-based analysis.** Of the 328 aptamers in the Hypothesis-based analysis, 23 had statistically significant differences between DD and control, with nine overexpressed and fourteen underexpressed after adjusting for multiple comparisons. Nine of these genes appeared in hypothesis-free and hypothesis-based analyses, as indicated by *. Plain italics indicate an indirect DD relationship, if any, to the protein; bold italics indicate a direct relationship between published DD findings and the protein. In the Notes column, "Referred to" means the referenced publication referred to the gene by an alias name. Exp: DD expression compared to controls. FDR: False Discovery Rate (Adjusted p-value) of expression difference. Nodes: number of pathway analysis connections (nodes) of this protein to others in this group with a medium confidence interaction score (0.400) and an FDR<=0.05. Notes: potential relationship to DD biology. Cat: categories of possible relationships to DD: 1. Apoptosis and senescence (2 genes); 2. Bioinformatics (7 genes); 3. Clinical and demographic (3 genes); 4. Extracellular matrix (9 genes); 5. Fibroblast and myofibroblast cytoskeleton, membrane, and motility (6 genes); 6. Fibroblast and myofibroblast differentiation and transcriptome (5 genes); 7. Vascular and perivascular (4 genes).
(DOCX)

**S5 Table. Functionally enriched pathways derived from the 23 differentially expressed proteins in the hypothesis-based analysis.** ID: Enriched category name. STR: Strength of protein-protein interaction (PPI). FDR: False Discovery Rate (Adjusted p-value). Overall PPI enrichment p-value: 1.35E-07. Hypothesis-based candidate protein preselection may influence enrichment results. Figure 6 summarizes enriched pathways for individual genes.
(DOCX)

**S6 Table. Genes referenced in the manuscript and their aliases.** This table lists all gene names referenced in the manuscript, their UniProt identifiers, protein names, and gene aliases.
(DOCX)

**S7 Table. Enriched pathways in Mass Spec – SomaScan concordant genes.** Six proteins identified by Mass Spectrometry had statistically significant differences between Dupuytren and control cohorts in the SomaScan analysis, representing genes AOC3, POSTN, SERPINC1, TF, PCSK9, and KNG1. Within this network, AOC3, POSTN, and PCSK9 had strong pathway connections, and PCSK9, SERPINC1, TF, and KNG1 had functionally enriched pathways. PPI enrichment p-value: 0.00987. STR: Strength of association. FDR: False Discovery Rate (adjusted p-value).
(DOCX)

## Acknowledgments

Gail Solomon assisted with manuscript writing.

Databank and study support from FORWARD, the National Databank for Rheumatic Diseases, is gratefully acknowledged.

## Author contributions

**Conceptualization:** Charles Eaton.

**Data curation:** Blake Hummer, Charles Eaton.

**Formal analysis:** Blake Hummer, Paola Sebastiani, Anastasia Leshchyk, Anastasia Gurinovich, Cecilie Bager, Morten Karsdal, Signe Nielsen, Charles Eaton.

**Funding acquisition:** Charles Eaton.

**Investigation:** Blake Hummer, Charles Eaton.

**Methodology:** Blake Hummer, Paola Sebastiani, Morten Karsdal, Charles Eaton.

**Project administration:** Charles Eaton.

**Resources:** Blake Hummer, Charles Eaton.

**Software:** Blake Hummer.

**Supervision:** Paola Sebastiani, Morten Karsdal, Charles Eaton.

**Validation:** Blake Hummer, Paola Sebastiani.

**Visualization:** Blake Hummer, Paola Sebastiani, Anastasia Leshchyk, Anastasia Gurinovich, Charles Eaton.

**Writing – original draft:** Blake Hummer, Paola Sebastiani, Charles Eaton.

**Writing – review & editing:** Blake Hummer, Paola Sebastiani, Charles Eaton.

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
