## [Decision Letter · Decision Letter 0]

24 Nov 2025

Dear Dr. Eaton,

Thank you for submitting your manuscript to PLOS ONE. After careful consideration, we feel that it has merit but does not fully meet PLOS ONE’s publication criteria as it currently stands. Therefore, we invite you to submit a revised version of the manuscript that addresses the points raised during the review process.

We look forward to receiving your revised manuscript.

Kind regards,

Luis A Arráez-Aybar

Academic Editor

PLOS ONE

Journal Requirements:

“I have read the journal's policy and the authors of this manuscript have the following competing interests: Morten Karsdal, Cecilie Bager, and Signe Holm Nielsen are all currently employed by Nordic Bioscience; Morten Karsdal, Cecilie Bager, and Signe Holm Nielsen hold stocks in Nordic Bioscience. The rest of the authors have no conflicts of interest to declare.”

We note that one or more of the authors are employed by a commercial company: Nordic Bioscience; Morten Karsdal, Cecilie Bager, and Signe Holm Nielsen

Additional Editor comments:

The decision on your submission is Minor Revision.

Reviewer 1 commended the technical rigor, clarity, and thoroughness of your analyses, highlighting that your integration of hypothesis-free and hypothesis-driven proteomic approaches represents an important advance in this field. They recommended one minor textual adjustment: reconsidering or removing the rationale about the correlation between tissue findings and circulating biomarkers, as this may not be necessary to justify your approach.

Reviewer 2 also found the work interesting and relevant, noting its potential significance for the biological understanding of Dupuytren’s disease. Their comments focus mainly on wording and clarity in the Introduction and Methods sections—for example, clarifying statements regarding treatment complication rates, revising the description of skin grafting, avoiding abbreviations such as “don’t,” and rephrasing certain sentences for precision and consistency.

Reviewers' comments:

Reviewer's Responses to Questions

**Comments to the Author**

1. Is the manuscript technically sound, and do the data support the conclusions?

Reviewer #1: Yes

Reviewer #2: Yes

2. Has the statistical analysis been performed appropriately and rigorously?

Reviewer #1: Yes

Reviewer #2: Yes

3. Have the authors made all data underlying the findings in their manuscript fully available?

Reviewer #1: Yes

Reviewer #2: Yes

4. Is the manuscript presented in an intelligible fashion and written in standard English?

Reviewer #1: Yes

Reviewer #2: Yes

Reviewer #1: This is a very interesting report describing the authors’ attempt to create “a staging biomarker profile to develop preventive therapeutics to improve long-term outcomes”. Toward this goal, the authors performed a DD blood proteomic biomarker profile by comparing DD plasma to healthy control samples. Both hypothesis-free and hypothesis-driven methodologies were employed, and the authors are to be commended on the thoroughness of their analyses. The authors have consistently provided potential links between their findings and previous reports without over-interpretation. The report provides a well-balanced and thorough overview of the current state of knowledge in this disease.

I did not detect any areas requiring major revisions. A minor point: the authors state that “DD-related tissue findings do not always correlate with blood measurements, and DD tissue gene expression does not correlate with local protein levels. Blood remains the most promising source for a disease activity profile.” I’m unaware that tissue findings, such as local gene expression and protein levels, and blood measurements frequently correlate in any fibrotic disease, especially tissue-type localised ones such as keloid. They often don’t correlate in most cancers either, where biomarkers are consistently utilized as surrogate indicators of disease staging. I’m sure there are plenty of gene transcripts and proteins that are modified in the local Dupuytren Disease/Contracture microenvironment that are not reflected in circulating blood samples. That the collection of samples from the disease environment is associated with a risk of disease exacerbation, and that there are currently no reliable biomarkers for this disease, are surely more than sufficient justifications for a study designed to detect novel circulating biomarkers. The authors could consider removing the (in my opinion, unnecessary) tissue correlation rationale from the document.

The findings concerning type-I collagen degradation versus production are intriguing and timely, especially considering the parallel reports noted by the authors of SNPs in genes encoding matrix metalloproteases in those with Dupuytren Disease that modify function. Similarly, the identification of changes in OSCAR and SPART expression levels are intriguing and the speculations by the authors correlating their known functions with Dupuytren Disease pathophysiology are plausible.

I was also intrigued by potential correlations between collagen glycosylation (glycation)-induced impaired type-I collagen degradation and the prevalence of Dupuytren Disease/Contracture in people with diabetes. The data reported here may also have broader implications for sclerotic changes in diabetes.

In summary, this a technically solid, well written, and thorough study with appropriate statistical analyses that addresses an important clinical problem, the identification of biomarkers for individuals in need of preventative interventions to minimise Dupuytren Disease recurrence.

Reviewer #2: Strong Points

This is an interesting paper. The future of successful treatment of Dupuytren’s will be biology rather than surgery so this sort of paper is an important basis

• Line 22- the statement that common treatments have high rates of complications is not quite correct. PNF has a low rate of complications.

• Line 51: full thickness skin grafting is not palliative, it is usually curative.

• Line 65: use the full “do not” rather than the abbreviation “don’t”

• Line 70: what is a “discovery study”- is this not a tautology?

• Line 70-74 contains no reference to a previous study and so it appears to be an abstract or summary of the current study, more suited to the abstract or a conclusion at the end of the paper.

• Line 91: sex is of course a biological variable. Why would the Authors assume that it was not relevant – their study could show this one way or another. The Authors might wish to explain to the Reader who might inadvertently wonder if this is not an unvalidated woke assumption…

**Do you want your identity to be public for this peer review?** For information about this choice, including consent withdrawal, please see our Privacy Policy

Reviewer #1: No

Reviewer #2: No

---

## [Author Response · Author response to Decision Letter 1]

22 Jan 2026

Editor

1. Please provide an amended Funding Statement declaring this commercial affiliation, as well as a statement regarding the Role of Funders in your study.

Response: Amended Funding Statement: The funder, Nordic Bioscience, provided support in the form of salaries for MK, CB, and SHN. Nordic Bioscience did not have any additional role in the study design, data collection and analysis, decision to publish, or preparation of the manuscript. The specific roles of these authors are articulated in the Author Contributions section. PS, AL, and AG are recipients of grant UM1TR004398 (the Tufts NIH CTSA award).

Response: Amended Competing Interests Statement The authors declare the following competing interests: Morten Karsdal, Cecilie Bager, and Signe Holm Nielsen are employees of Nordic Bioscience and hold stocks in Nordic Bioscience. No other competing interests have been declared. This does not alter the authors’ adherence to PLOS ONE policies on sharing data and materials. The rest of the authors have no conflicts of interest to declare.

3. Please include captions for your Supporting Information files at the end of your manuscript, and update any in-text citations to match accordingly.

Response: Updated as recommended. References previously only in supporting information files have been added to the supporting information captions and merged into the primary bibliography.

4. If the reviewer comments include a recommendation to cite specific previously published works, please review and evaluate these publications to determine whether they are relevant and should be cited.

Response: Not Applicable.

5. If the reviewer comments include a recommendation to cite specific previously published works, please review and evaluate these publications to determine whether they are relevant and should be cited.

Response: Partially related to this: One reference was removed; see details in the response to reviewer 1.

6. If applicable, we recommend that you deposit your laboratory protocols in protocols.io to enhance the reproducibility of your results.

Response: Not applicable. For clarity, I added "No laboratory protocols were conducted in-house" to the Assays section.

Response: I updated the style and file names to the requested specs.

Reviewer 1

1. A minor point: the authors state that "DD-related tissue findings do not always correlate with blood measurements, and DD tissue gene expression does not correlate with local protein levels. Blood remains the most promising source for a disease activity profile." I'm unaware that tissue findings, such as local gene expression and protein levels, and blood measurements frequently correlate in any fibrotic disease, especially tissue-type localised ones such as keloid. They often don't correlate in most cancers either, where biomarkers are consistently utilized as surrogate indicators of disease staging. I'm sure there are plenty of gene transcripts and proteins that are modified in the local Dupuytren Disease/Contracture microenvironment that are not reflected in circulating blood samples. That the collection of samples from the disease environment is associated with a risk of disease exacerbation, and that there are currently no reliable biomarkers for this disease, are surely more than sufficient justifications for a study designed to detect novel circulating biomarkers. The authors could consider removing the (in my opinion, unnecessary) tissue correlation rationale from the document.

Response: Agree, scientist readers should know that neither tissue nor blood findings necessarily correlate, nor do gene expression and protein measurements. However, this article's target audience includes hand surgeons in private practice with a different knowledge base, and I had included this sentence for the benefit of these interested clinician readers. Action: "DD-related tissue findings do not always correlate with blood measurements, and DD tissue gene expression does not correlate with local protein levels." was removed, along with the citation unique to this sentence (19.Jupp O, Pullinger M, Marjoram T, Lott M, Chojnowski AJ, Clark IM. Biomarkers of Postsurgical Outcome in Dupuytren Disease. Dupuytren Disease and Related Diseases - The Cutting Edge 2017. p. 55-61.)

Reviewer 2.

1. Line 22- the statement that common treatments have high rates of complications is not quite correct. PNF has a low rate of complications.

Response: I agree. The Devil's dilemma in Dupuytren is the choice between procedures with high complication rates or those with high early recurrence rates.

Original: “The most common treatments have high rates of complications and early recurrence.”

Updated: “The most common treatments have either high complication rates or high early recurrence rates.”

2. Line 51: full thickness skin grafting is not palliative, it is usually curative.

Response: Skin grafting, per se, does not change recurrence rates per Ullah 2009 https://boneandjoint.org.uk/Article/10.1302/0301-620X.91B3.21054. Although textbook dermofasciectomy (radical skin excision and resurfacing with large skin grafts or flaps) has very low recurrence rates, it is not commonly performed compared to fasciectomy +/- skin graft, collagenase, or PNF.

Original: "Current treatments are palliative, often with partial or temporary improvement."

Updated: "Common current treatments are palliative, often with partial or temporary improvement."

3. Line 65: use the full "do not" rather than the abbreviation "don't"

Response: Original: "don’t”

Updated: “do not”

4. Line 70: what is a “discovery study”- is this not a tautology?

Response: The word “discovery” was intended to indicate that the study’s objective was to acquire preliminary data in an uncharted area and distinguish it from research aiming to extend or validate an area of established knowledge.

Updated: Lines 70-74 were removed.

5. Line 70-74 contains no reference to a previous study and so it appears to be an abstract or summary of the current study, more suited to the abstract or a conclusion at the end of the paper.

Response: Updated: Lines 70-74 were removed.

6. Line 91: sex is of course a biological variable. Why would the Authors assume that it was not relevant – their study could show this one way or another. The Authors might wish to explain to the Reader who might inadvertently wonder if this is not an unvalidated woke assumption…

Response: Original: “Sex was not considered a biological variable.”

Updated: ”Sex was considered a biological variable. While we controlled for sex differences, we did not stratify by sex due to limited sample sizes.”

---

## [Editor Report · Decision Letter 1]

11 Feb 2026

Identification of novel plasma proteomic biomarkers of Dupuytren Disease

PONE-D-25-45067R1

Dear Dr. Eaton,

We’re pleased to inform you that your manuscript has been judged scientifically suitable for publication and will be formally accepted for publication once it meets all outstanding technical requirements.

Kind regards,

Luis A Arráez-Aybar

Academic Editor

PLOS One
---

## [Editor Report · Acceptance letter]

PONE-D-25-45067R1

PLOS One

Dear Dr. Eaton,

I'm pleased to inform you that your manuscript has been deemed suitable for publication in PLOS One. Congratulations! Your manuscript is now being handed over to our production team.

Kind regards,

on behalf of

Dr. Luis A Arráez-Aybar

Academic Editor

PLOS One